 **eLIFE**

# Functional genomic characterization of neoblast-like stem cells in larval *Schistosoma mansoni*

Bo Wang[1,2], James J Collins III[1,3], Phillip A Newmark[1,2,3]*

[1]Howard Hughes Medical Institute, University of Illinois at Urbana-Champaign, Urbana, United States; [2]Institute for Genomic Biology, University of Illinois at Urbana-Champaign, Urbana, United States; [3]Department of Cell and Developmental Biology, University of Illinois at Urbana-Champaign, Urbana, United States

**Abstract** Schistosomes infect hundreds of millions of people in the developing world. Transmission of these parasites relies on a stem cell-driven, clonal expansion of larvae inside a molluscan intermediate host. How this novel asexual reproductive strategy relates to current models of stem cell maintenance and germline specification is unclear. Here, we demonstrate that this proliferative larval cell population (germinal cells) shares some molecular signatures with stem cells from diverse organisms, in particular neoblasts of planarians (free-living relatives of schistosomes). We identify two distinct germinal cell lineages that differ in their proliferation kinetics and expression of a *nanos* ortholog. We show that a *vasa*/PL10 homolog is required for proliferation and maintenance of both populations, whereas *argonaute2* and a fibroblast growth factor receptor-encoding gene are required only for *nanos*-negative cells. Our results suggest that an ancient stem cell-based developmental program may have enabled the evolution of the complex life cycle of parasitic flatworms.

**\*For correspondence:** pnewmark@life.illinois.edu

**Competing interests:** The authors declare that no competing interests exist.

**Reviewing editor**: Alejandro Sánchez Alvarado, Howard Hughes Medical Institute, Stowers Institute for Medical Research, United States

## Introduction

*Schistosoma* flatworms infect 230 million people worldwide and cause ~250,000 deaths per year (***van der Werf et al., 2003***). These trematodes are transmitted through a life cycle that alternates between asexual and sexual generations in invertebrate intermediate and vertebrate definitive hosts, respectively (***Clark, 1974***; ***Shoop, 1988***). The life cycle initiates as eggs are excreted from a mammalian host into freshwater, releasing ciliated, free-swimming larvae called miracidia that seek out and penetrate a snail intermediate host. Entry into the snail triggers a series of morphological, physiological, and biochemical transformations (***Basch and DiConza, 1974***; ***Kawamoto et al., 1989***; ***Ludtmann et al., 2009***; ***Wu et al., 2009***; ***Parker-Manuel et al., 2011***), followed by a clonal expansion of the larvae (called sporocysts at this stage) inside the snail host, ultimately producing thousands of infective cercariae (***Figure 1A***) (***Cheng and Bier, 1972***; ***Ward et al., 1988***). Mature cercariae then emerge from the snail into freshwater, burrow through the epidermis of mammalian hosts, migrate to species-specific niches in the host vascular system, develop to adulthood, and begin to reproduce sexually, thereby completing the life cycle. Thus, asexual amplification inside of the snail is vital for propagation of schistosomes.

A population of totipotent stem cells, historically called 'germinal cells', is thought to underlie this unique intramolluscan amplification by undergoing multiple rounds of proliferation and de novo embryogenesis in the absence of fertilization (***Olivier and Mao, 1949***; ***Cort et al., 1954***; ***Whitfield and Evans, 1983***). Early ultrastructural and histological studies recognized these cells by their stem cell-like morphology and rapid cycling kinetics (***Schutte, 1974***; ***Pan, 1980***). In support of the totipotency of these germinal cells, serial transplantation of sporocysts into naive snail hosts led to continuous sporocyst propagation and cercarial production (***Jourdane and Théron, 1980***). These classic studies

**eLife digest** Schistosomiasis—a disease caused by parasitic flatworms known as schistosomes—affects more than 200 million people worldwide, mainly in tropical regions, and in public health importance is second only to malaria (according to the World Health Organization). Chronic infection leads to damage to internal organs, and the disease is responsible for roughly 250,000 deaths each year.

The schistosome parasite has a complex life cycle, and the worms are capable of infecting mammals during just one stage of this cycle. Infection occurs through contact with contaminated freshwater, with the infectious form of the parasite burrowing through skin. Once inside the body, the parasites mature into adults, before reproducing sexually and laying eggs that are excreted by their host back into the water supply.

However, to generate the form of the parasite that can infect mammals, schistosomes must first infect an intermediate host, namely a freshwater snail. When the larval form of the parasite—which cannot infect mammals—enters the snail, the larvae undergo an unusual type of asexual embryogenesis. This results in thousands of parasites that are capable of infecting mammals. Studies suggest that a population of cells known as germinal cells are responsible for this transformation and replication process, but little is known about these cells at the molecular level.

Here, Wang et al. report the gene expression profile of these cells in a species of schistosome, and use RNA-mediated silencing techniques to explore the functions of the genes. This analysis revealed that the germinal cells have a molecular signature similar to that of neoblasts—adult pluripotent stem cells found in free-living flatworms such as planarians. Neoblasts can develop into any cell type in the body, enabling planarians to repair or even replace damaged body parts.

The similarity between neoblasts and germinal cells led Wang et al. to suggest that schistosomes may have evolved their parasitic life cycle partly by adapting a program of development based on stem cells in non-parasitic worms.

led to the model that division of these diploid presumptive totipotent stem cells in mother sporocysts produces progeny that are able to independently initiate the embryogenesis of daughter sporocysts (*Whitfield and Evans, 1983*). These daughter sporocysts, which are essentially sacs filled with germinal cells, can then produce more daughter sporocysts or infective cercariae in the same manner as they were generated themselves. This process represents 'polyembryony'—during which multiple embryos are produced from the same zygote with no intervening gamete production. Thus, germinal cells appear to possess a unique developmental program, and it is unknown how they are specified, maintained, and regulated molecularly.

In planarians, free-living flatworm relatives of schistosomes, a population of pluripotent stem cells called neoblasts can regenerate injured tissues and replenish a whole animal from a single cell (*Newmark and Sánchez Alvarado, 2002*; *Wagner et al., 2011*). Guided by this knowledge, we recently identified a population of neoblast-like cells in adult *Schistosoma mansoni* (*Collins et al., 2013*). These observations led us to hypothesize that germinal cells underlying schistosome asexual amplification may share a similar molecular program. Here, we show that the proliferating cells in sporocysts express many conserved stem cell genes. Using RNA interference (RNAi) we identify conserved regulators that are required to maintain the proliferative capacity of these cells. The similarity between these germinal cells in schistosome larvae, somatic stem cells in schistosome adults, and planarian neoblasts, links embryonic development and homeostatic tissue maintenance in these parasites; furthermore, it suggests that adaptation of an ancient stem cell developmental program may have enabled the evolution of complex trematode life cycles.

## Results

### Tracking germinal cells through the intramolluscan stages of the schistosome life cycle

Based on previous studies showing that germinal cells in schistosome larvae have a distinct morphology—high nuclear-to-cytoplasmic ratio, a large nucleolus, and cytoplasm densely packed with ribosomes (*Pan, 1980*)—we reasoned that nucleic acid stains that preferentially label RNA could provide a means to

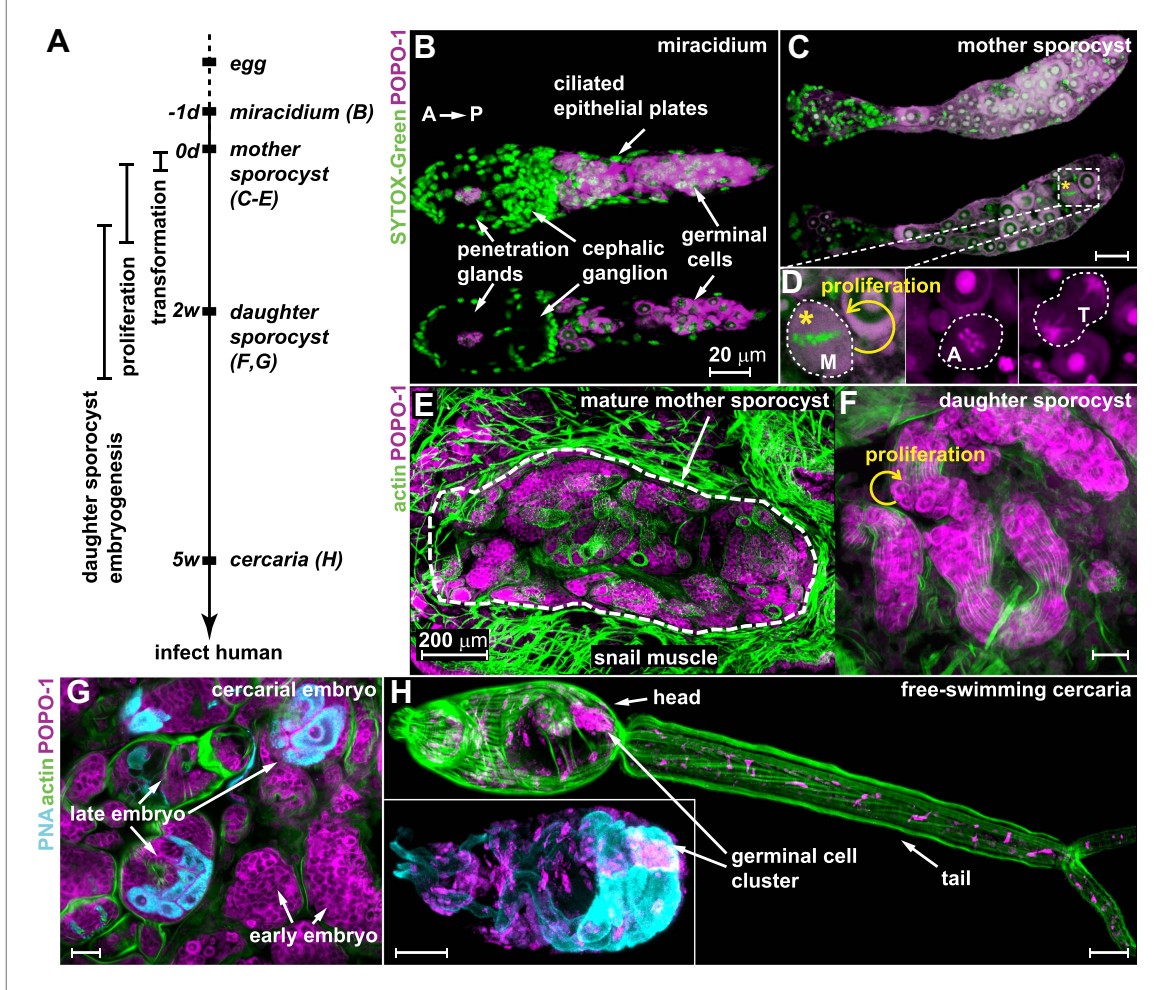

**Figure 1**. Germinal cells are detected throughout the asexual phase of the *S. mansoni* life cycle. (**A**) A schematic timeline of schistosome asexual amplification. (**B–C**) Maximum intensity projections of confocal stacks (top) and single optical slices (bottom) of a POPO-1 and SYTOX-Green co-stained miracidium (**B**) and a sporocyst 24 hr after in vitro transformation (**C**). (**D**) Representative images of cells at metaphase (M), anaphase (A), and telophase (T) (from left to right), captured in sporocysts 24 hr post-transformation. (**E–G**) Cryosections of the tentacle of a *Biomphalaria glabrata* snail showing a mother sporocyst (perimeter highlighted by dashed line) with daughter sporocysts packed inside (3 weeks post infection) (**E**); an individual daughter sporocyst that has migrated to the digestive glands of a *B. glabrata* snail 6 weeks post infection (**F**); and cercarial embryos within a daughter sporocyst in the digestive glands of a *B. glabrata* snail 6 weeks post infection (**G**) (staged after ***Cheng and Bier, 1972***). Actin is stained with phalloidin. Peanut agglutinin (PNA) visualizes acetabular glands and ducts of the cercariae. (**H**) A mature cercaria. The inset shows a magnified view of this animal's head visualized with PNA and POPO-1 staining. Scale bars are 20 μm, except in (**E**) which is 200 μm.

label these cells specifically. Thus, we screened a number of nucleic acid-binding dyes and determined that POPO-1 clearly distinguishes the RNA-rich germinal cells from the other somatic cell types. In particular, POPO-1 strongly stains the nucleolus and cytoplasm of the germinal cells. This staining enabled us to track these cells through various stages of intramolluscan development both in vitro and in vivo (***Figure 1***).

Consistent with previous work (***Pan, 1980***), we found 10–20 germinal cells in the posterior half of the body of free-swimming miracidia after hatching from the egg (***Figure 1B***). In vitro transformation of miracidia into sporocysts triggered germinal cell proliferation (***Yoshino and Laursen, 1995***; ***Ivanchenko et al., 1999***; ***Bixler et al., 2001***): we observed these cells in mitosis as early as 24 hr post-transformation (~0.2 mitoses per animal) (***Figure 1C,D***), which is consistent with the behavior of these cells in vivo after miracidia penetrate a snail host (***Schutte, 1974***). Following their long-term development in vivo, POPO-1 staining identifies a similar cell type throughout various stages of schistosome asexual development, including: developing daughter sporocysts in the mother (***Figure 1E***); motile daughters that have broken away from the mother and migrated to the snail digestive glands

(*Figure 1F*); and cercarial embryos developing inside the daughter sporocysts (*Figure 1G*). These POPO-1-labeled cells ultimately segregated into a compact cluster in cercariae (*Cheng and Bier, 1972*) (*Figure 1H*), presumably saved for the next stage of development after their penetration into mammalian hosts (*Dorsey et al., 2002*). These observations reveal a morphologically homogeneous presumptive stem cell population that persists through the larval stages of the schistosome life cycle.

## Proliferation kinetics at the miracidium-to-sporocyst transition

In order to characterize these cells molecularly, we focused on the miracidium-to-mother sporocyst transition. To define this transition more precisely and examine the initiation of cell proliferation, at different times following in vitro transformation, we treated sporocysts with 5-ethynyl-2'-deoxyuridine (EdU), a thymidine analogue that is incorporated into DNA during S-phase of the cell cycle (*Salic and Mitchison, 2008*). For these experiments, miracidia were transformed to mother sporocysts in vitro and co-cultured with an immortalized snail cell line (*Bge* cells) under hypoxic conditions (*Figure 2A*) (*Yoshino and Laursen, 1995*; *Ivanchenko et al., 1999*; *Bixler et al., 2001*). During the first 20 hr after transformation, few cells incorporated EdU. However, between 20–24 hr, we typically observed 1–2 clusters of EdU$^+$ germinal cells, which are identified by their nucleolar and cytoplasmic POPO-1 staining. At later time points (48–64 hr), a large fraction of germinal cells in the sporocyst incorporated EdU, and all EdU$^+$ cells exhibit a germinal cell-specific morphology. These cells were actively dividing as we routinely observed many mitotic figures in these samples (*Figures 1D and 2A*). As a result of this massive

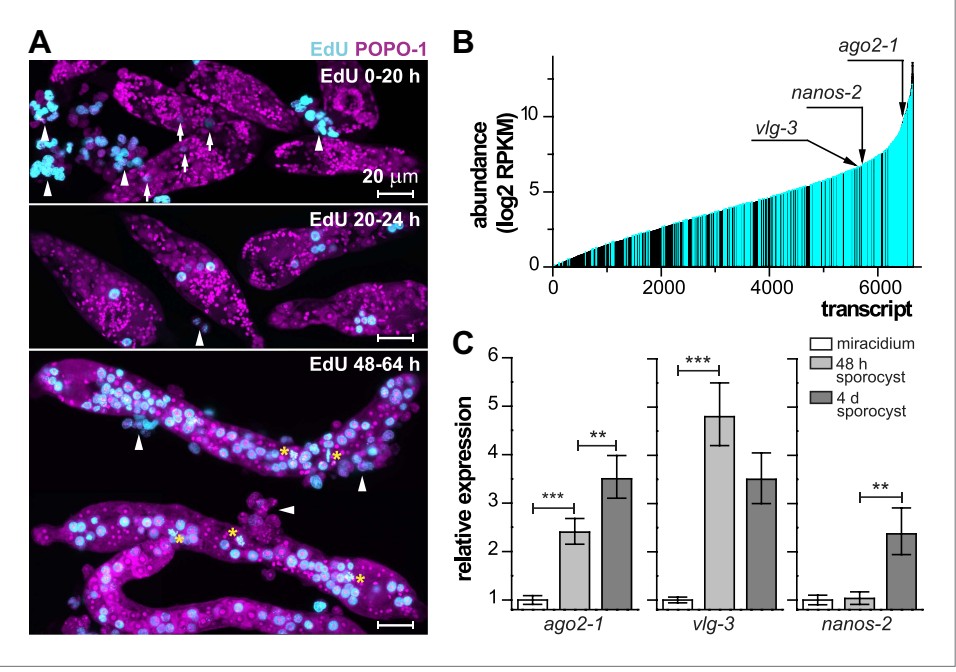

**Figure 2**. Life-cycle stage-specific transcriptional profiling to characterize germinal cell gene expression. (**A**) EdU labeling to detect proliferating cells at various time points following transformation. For these experiments sporocysts were co-cultured with *Bge* cells, a cell line derived from embryos of *B. glabrata* snails, to sustain the normal development of larval schistosomes. Arrows indicate cells weakly incorporating EdU at the early time point; arrowheads indicate proliferating *Bge* cells. Asterisks highlight cells in mitosis. In the absence of SYTOX-Green, POPO-1 also stains somatic nuclei, which are small and compact. Scale bars are 20 µm. (**B**) Transcripts expressed in sporocysts 48 hr post-transformation, ranked by abundance as measured by RNAseq. Orthologs shared between sporocyst-enriched genes and planarian neoblast-enriched transcripts are highlighted in cyan. (**C**) Relative expression levels of *ago2-1*, *vlg-3*, and *nanos-2* during sporocyst development with respect to the expression in miracidia, measured by qPCR in biological triplicate. Error bars are standard deviations. **p<0.01. ***p<0.001 (*t*-test).

The following figure supplements are available for figure 2:

**Figure supplement 1**. Transcriptional profiling reveals genome-wide similarity between schistosome germinal cells and planarian neoblasts.

proliferative burst, these germinal cells increased dramatically in number and came to occupy most of the sporocyst body, tripling sporocyst size during the first 3 days. These results suggest that a tightly regulated, sharp developmental transition begins at ~20 hr post-transformation, when the first wave of proliferation starts.

## Transcriptional profiling identifies germinal cell-associated transcripts

Since the miracidium-to-sporocyst transition is associated with both an increase in the number of proliferative germinal cells and a switch from a quiescent state to an actively proliferating state, we reasoned that the transcriptional profile of sporocysts would be enriched for mRNAs associated with germinal cells. Thus, we compared the gene expression profiles of miracidia and sporocysts 48 hr post-transformation using RNA sequencing (RNAseq). Of the 10,852 predicted genes in the annotated *S. mansoni* genome (*Berriman et al., 2009*; *Protasio et al., 2012*), 6677 genes were detected in 48 hr sporocysts with an RPKM (reads per kilobases of transcript per million mapped reads) value above 1 (*Figure 2B*); 1662 of these genes were upregulated relative to miracidia (*Figure 2—figure supplement 1A* and *Supplementary file 1A*).

Besides the clonal expansion of germinal cells, the transition from miracidium-to-sporocyst is associated with numerous physiological and anatomical changes. Reasoning that germinal cells and planarian neoblasts may share a similar transcriptional signature, we compared our dataset with transcripts enriched in FACS-purified planarian neoblasts (*Önal et al., 2012*) to identify putative germinal cell-specific transcripts. Indeed, we noticed substantial overlap between these two datasets: of the 1662 genes upregulated in sporocysts, ~30% of them (581 genes) shared similarity (e-value < $e^{-10}$) with neoblast-enriched transcripts (*Figure 2—figure supplement 1*, and *Supplementary file 1C*). Conversely, more than ~20% (864/4032) of neoblast-enriched transcripts shared similarity with sporocyst-enriched mRNAs. To better define the similarity between these datasets, we identified a total of 4749 1:1 schistosome-to-planarian orthologs using reciprocal BLAST comparisons (e-value < $e^{-10}$) (*Figure 2B*). 1579 orthologs were enriched in planarian neoblasts; quite interestingly, 1525 of them (96.5%) were expressed in sporocysts as well (RPKM in sporocysts >1). We examined the overlap between neoblast-enriched transcripts and sporocyst-enriched genes and identified 331 orthologs (20% of 1662 genes that are upregulated in sporocysts) (*Figure 2B*, and *Supplementary file 1D*). This list may contain a core set of genes that are essential for stem cell proliferation and maintenance. As a comparison, 12% (1248/10,125) of the remaining schistosome transcripts have orthologs in the planarian neoblast transcriptome. Given the differences in how these datasets were generated (i.e., FACS-purified cells from planarian vs whole schistosome sporocysts), together with the fact that the miracidium-to-sporocyst transformation is expected to result in transcriptional changes independent of germinal cell number, this degree of overlap seems likely to underestimate the similarity between the transcriptomes of germinal cells and planarian neoblasts.

In addition to components of cell cycle and DNA repair machinery, examination of these overlapping gene sets identified many conserved factors associated with stem cell maintenance and germ cell development in diverse organisms (*Table 1*), including a pair of fibroblast growth factor receptors (*fgfr*) (*Ogawa et al., 2002*; *Lanner and Rossant, 2010*; *Wagner et al., 2012*; *Collins et al., 2013*), three components of PRC2 (polycomb repressive complex 2; suppressor of Zeste 12, *Sm-sz12*, enhancer of Zeste, *Sm-ezh*, and embryonic ectoderm development, *Sm-eed*) (*Surface et al., 2010*; *Wagner et al., 2012*; *Önal et al., 2012*), a p53 homolog (*Pearson and Sánchez Alvarado, 2010*), a *bruno*-like (*bruli*) RNA-binding protein (*Guo et al., 2006*), and an *argonaute2*-like gene (*Gomes et al., 2009*; *Rouhana et al., 2010*; *Li et al., 2011*; *Leonardo et al., 2012*). Strikingly, 11/37 DEAD-box helicases (DBHs) in the *S. mansoni* genome are upregulated in sporocysts (*Supplementary file 1B*), including three *vasa*/PL10 homologs (*Skinner et al., 2012*) that may play a similar role in flatworm parasites to that of *vasa* in other metazoans (*Juliano et al., 2010*; *Tsai et al., 2013*) (for consistency with previous studies [*Shibata et al., 1999*; *Ohashi et al., 2007*; *Skinner et al., 2012*], we will refer to these *vasa*/PL10 homologs as *vasa*-like genes [*vlg*]). Taken together, this comparison between parasitic and free-living flatworms suggests that proliferating germinal cells in sporocysts share some common features with planarian stem cells. Interestingly, this set of conserved germinal cell-enriched genes also contains several targets accessible with available small-molecule drugs (e.g., cyclophilin, adenosylhomocysteinase, transketolase) (*Crowther et al., 2010*), suggesting that these cells could serve as a potential vulnerable point to block schistosome propagation.

## Conserved post-transcriptional regulators define distinct populations of germinal cells

In addition to their widely appreciated roles in germ cell specification, maintenance, and differentiation, a network of conserved post-transcriptional regulators (e.g., *piwi/argonaute*, *vasa*, and *nanos*)

**Table 1.** Expression in miracidia and sporocysts of schistosome homologs of planarian neoblast-enriched transcripts, measured by RNAseq

| Gene name | RPKM (miracidia) | RPKM (sporocysts) | Fold change |
|---|---|---|---|
| **vasa-like** (*vlg*, **Smp_068440**, 154320, 033710) | 20.6/46.1/272.4 | 98.5/79.2/423.1 | 4.8/1.7/1.6 |
| **polo kinase (Smp_009600)** | 43.0 | 149.5 | 3.5 |
| **fgfr (Smp_175590**, 157300) | 3.5/3.0 | 10.2/6.4 | 2.9/2.1 |
| *sz12* (Smp_047720) | 5.6 | 16.0 | 2.8 |
| *bruli* (Smp_041350) | 5.5 | 15.3 | 2.8 |
| *Sedt8* (Smp_055310) | 1.9 | 4.8 | 2.6 |
| *egr* (Smp_094930) | 3.0 | 7.4 | 2.5 |
| **cyclin B (Smp_082490)** | 79.7 | 194.0 | 2.4 |
| *nlk* (Smp_074080) | 7.5 | 18.0 | 2.4 |
| **ago2-1 (Smp_179320)** | 258.9 | 537.5 | 2.1 |
| **PCNA (Smp_046500)** | 194.0 | 395.0 | 2.0 |
| *inx* (Smp_141290) | 102.6 | 203.7 | 2.0 |
| *ezh* (Smp_078900) | 8.1 | 15.5 | 1.9 |
| PHB (Smp_075210, 075940) | 222.0/197.7 | 421.0/290.8 | 1.9/1.5 |
| *pp32a* (Smp_010940) | 641.7 | 1176.3 | 1.8 |
| **H2A (Smp_086860)** | 240.3 | 437.1 | 1.8 |
| *THOC* (Smp_005260) | 219.6 | 397.3 | 1.8 |
| egfr (Smp_093930, 165470) | 6.8/7.0 | 12.1/10.0 | 1.8/1.4 |
| *CHD* (Smp_158050) | 39.6 | 68.6 | 1.7 |
| tudor-like (Smp_081570) | 222.8 | 367.8 | 1.7 |
| **H2B (Smp_108390)** | 124.6 | 206.5 | 1.7 |
| *ef-tu* (Smp_073500) | 151.6 | 243.6 | 1.6 |
| *HSP60* (Smp_008545) | 2051.5 | 3224.7 | 1.6 |
| *fhl* (Smp_048560) | 2.0 | 3.1 | 1.5 |
| *eed* (Smp_165220) | 32.7 | 46.3 | 1.4 |
| *junl* (Smp_067520) | 5.5 | 7.7 | 1.4 |

Expression of genes in bold is confirmed in this study with qPCR or FISH.

are associated with multipotency in many metazoans (*Juliano et al., 2010*). In diverse organisms, co-expression of these 'germline genes' specifies progenitor cells that can contribute to both soma and germline (*Juliano et al., 2010*). Though *piwi* and *vasa* appear to be absent from schistosome genomes (*Gomes et al., 2009*; *Collins et al., 2013*), the *S. mansoni* genome has three *argonaute* homologs, two *nanos* homologs (*Figure 3—figure supplement 1*), and three *vasa*/PL10 homologs (*vlg*) (*Skinner et al., 2012*; *Tsai et al., 2013*). These genes, with the exception of *Sm-nanos-1* (Smp_057740), are all expressed abundantly in sporocysts. Among them, *Sm-ago2-1*, *Sm-vlg-3*, and *Sm-nanos-2*, are upregulated in sporocysts, but with different kinetics (*Figure 2C* and *Table 1*; for clarity, the prefix 'Sm' will be omitted from gene names for the rest of the paper): *ago2-1* increases steadily in expression throughout early sporocyst development; *vlg-3* expression increases sharply following transformation, then plateaus; and *nanos-2* mRNA expression does not increase until 4 days post-transformation.

To examine whether these genes are expressed in germinal cells, we developed a method for whole-mount RNA fluorescence in situ hybridization (FISH) in sporocysts. Using this method, we confirmed that *ago2-1* transcripts were enriched in germinal cells (*Figure 3A*). These cells also expressed the cell cycle-regulated gene *PCNA* (*Figure 3A* top). Following a 24 hr EdU pulse, >95% (2178/2282) of EdU+ cells expressed *ago2-1* and ~90% (1626/1815) of the *ago2-1*+ cells incorporated EdU (*Figure 3A*, bottom). Similar observations were made with *vlg-3* (*Figure 3B*), suggesting that both *ago2*-1 and *vlg-3* are expressed preferentially in proliferative germinal cells.

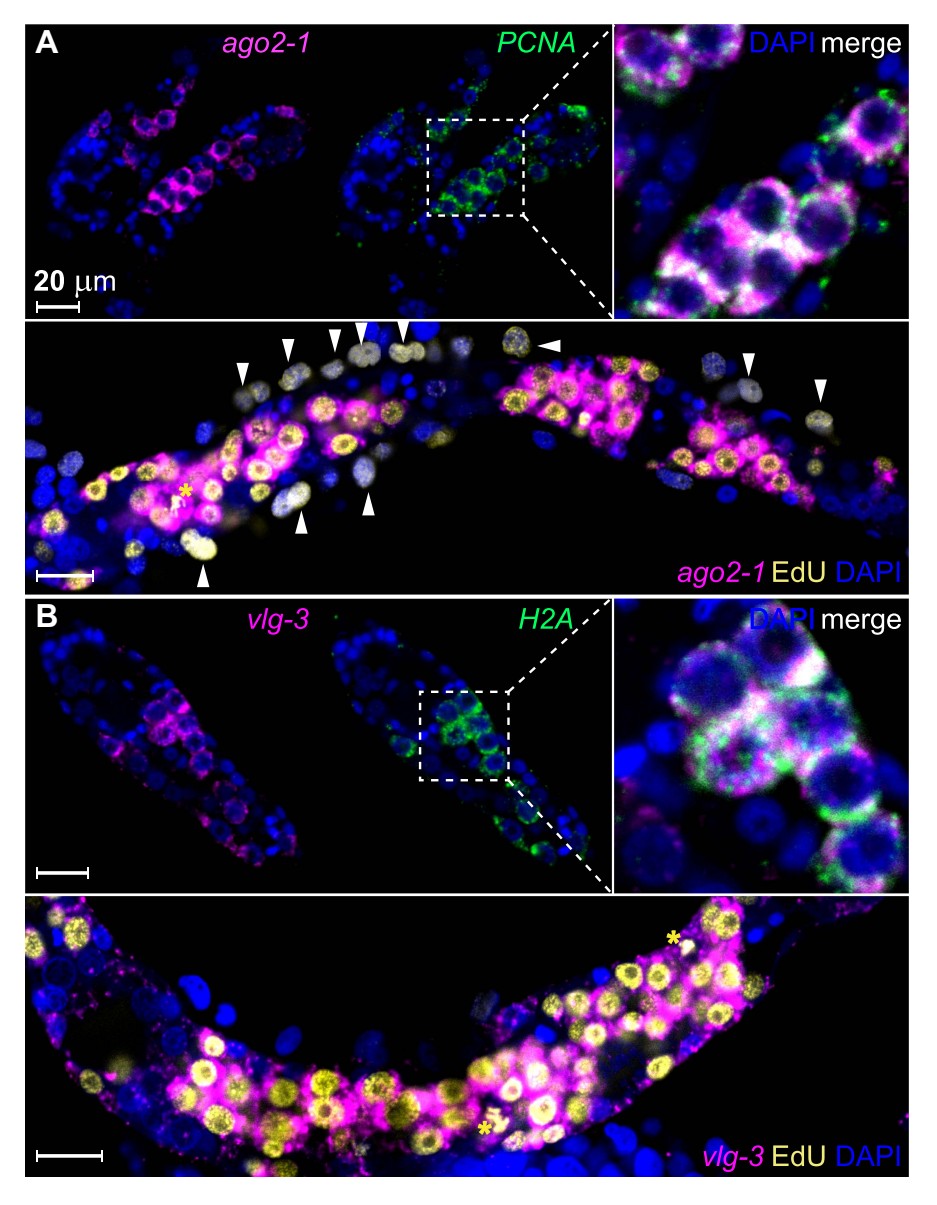

**Figure 3**. *ago2-1* and *vlg-3* are expressed in proliferative germinal cells. (**A**) Top: confocal sections showing colocalization of *ago2-1* and *PCNA* by FISH in sporocysts 24 hr post-transformation. Bottom: cells expressing *ago2-1* incorporate EdU after a pulse 48–72 hr post-transformation. (**B**) *vlg-3* is co-expressed with the cell cycle-associated transcript, *H2A* (top), and *vlg-3* is expressed in cells that incorporate EdU following a pulse at 48–72 hr post-transformation (1254 *vlg-3*+ EdU+/1362 EdU+ cells) (bottom). Arrowheads indicate proliferating Bge cells; asterisks indicate mitotic cells. Scale bars are 20 µm.

The following figure supplements are available for figure 3:

**Figure supplement 1**. SmAGO2-1 is homologous to PIWI and AGO2 proteins, and SmNANOS2 is homologous to NANOS proteins.

By contrast, *nanos-2* expression was detected only in a subset of *ago2-1*+/*vlg-3*+ cells (**Figure 4A,B**). This observation suggests the existence of two distinct populations of germinal cells: *nanos-2*+ or *nanos-2*– cells. Monitoring the ratio of these two cell populations over time, we found the *nanos-2*– population expanded at a higher rate (doubling time ~24 hr) than the *nanos-2*+ population (**Figure 4C**). To examine if these cells possess differences in their cell cycle kinetics, we pulsed sporocysts with EdU

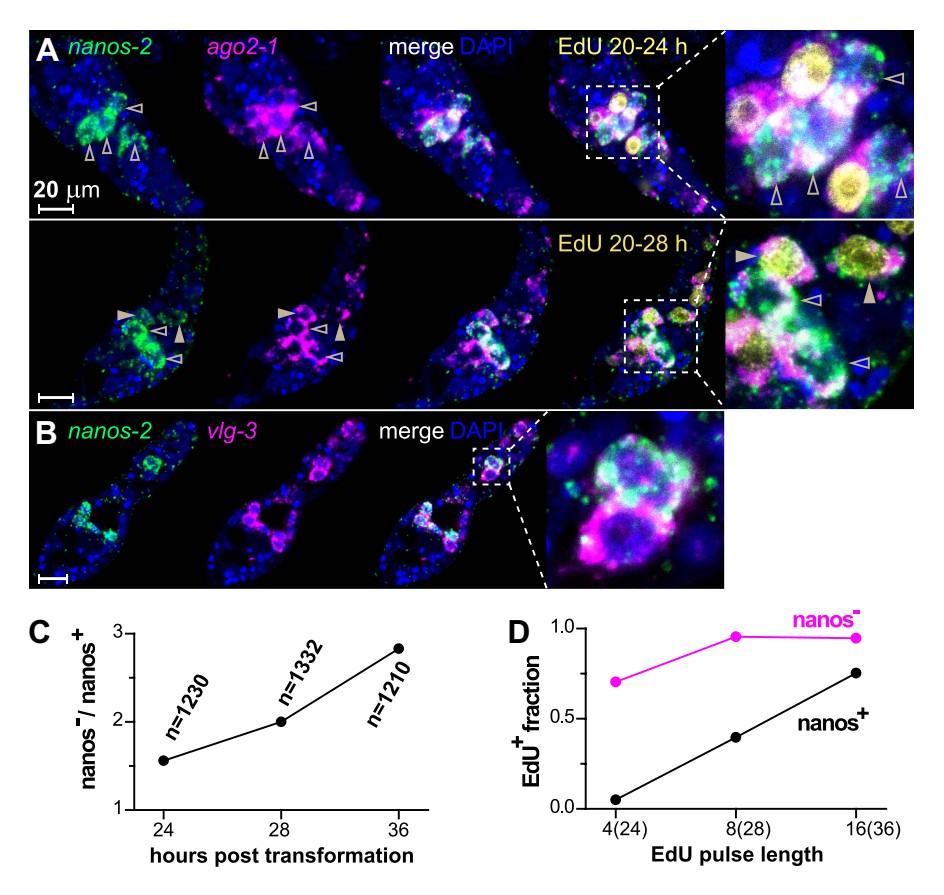

**Figure 4**. *ago2-1*, *vlg-3*, and *nanos-2* expression identifies heterogeneity in the germinal cell population. (**A**) FISH to detect *nanos-2* and *ago2-1* mRNA in EdU-labeled parasites. Relative to the *nanos-2⁻* germinal cells, *nanos-2⁺* cells require longer time periods to incorporate EdU. Germinal cells are defined as *ago2-1⁺* cells. The open arrowheads indicate *nanos-2⁺* cells that are EdU⁻, whereas filled arrowheads point to those that are EdU⁺. Times for EdU pulses are indicated in figures. (**B**) FISH to detect *nanos-2* and *vlg-3* mRNA. Scale bars are 20 μm. (**C**) The ratio between *nanos-2⁻* and *nanos-2⁺* germinal cells increases with time after transformation. (**D**) Fractions of cells that incorporate EdU after pulses of various lengths. All EdU pulses start at 20 hr post-transformation, and the end times are indicated in the parentheses along the x-axis.

for various lengths of time and determined the fraction of *nanos-2⁺* or *nanos-2⁻* germinal cells that were EdU⁺ (**Figure 4A,D**). Following a 4-hr EdU pulse, ~70% of the *nanos-2⁻* cells were EdU⁺, whereas only ~5% of the *nanos-2⁺* cells were EdU⁺. The fraction of *nanos-2⁺* cells that were EdU⁺ increased to ~40% with an 8-hr pulse, and to ~75% with a 16-hr pulse. With either an 8-hr or 16-hr pulse, ~95% of *nanos-2⁻* germinal cells were EdU⁺. To rule out the possibility that EdU⁺/*nanos-2⁺* cells are progeny of the more rapidly proliferating *nanos-2⁻* cells, we administrated a short 4-hr EdU pulse followed by a 12-hr chase period in the absence of EdU. We found that only 6% of *nanos-2⁺* cells were EdU⁺. Since a similar fraction of EdU⁺/*nanos-2⁺* cells is observed immediately following a pulse (**Figure 4D**), it is unlikely that *nanos-2⁻* germinal cells differentiate to produce *nanos-2⁺* cells. Collectively, these observations suggest that *nanos-2⁺* germinal cells have a longer cell cycle and enter S-phase less often than do *nanos-2⁻* germinal cells. This differential rate of proliferation likely explains why the upregulation of *nanos-2* expression is delayed following transformation (**Figure 2C**).

## *vasa* and *argonaute* homologs are required for germinal cell maintenance and proliferation

Motivated by the pivotal role of post-transcriptional regulation in various stem cell populations and during embryonic development (**Juliano et al., 2010**), we characterized the functions of *ago2-1* and

*vlg-3* using RNAi (**Boyle et al., 2003**; **Rinaldi et al., 2009**). For these experiments parasites were treated with double-stranded RNA (dsRNA) continuously while in the egg, after hatching, and during sporocyst development (**Figure 5A**). Using this procedure we were able to achieve robust reductions (>80%) in both *ago-2* and *vlg-3* mRNA levels, as measured by quantitative real-time PCR (qPCR). To assess the functions of these genes we monitored EdU incorporation in RNAi-treated parasites. Consistent with these genes regulating germinal cell proliferation, RNAi of either *ago2-1* or *vlg-3* resulted in significantly fewer EdU+ germinal cells following a 24-hr EdU pulse (**Figure 5B,C**). Inhibition of either *ago-2* or *vlg-3* also resulted in a significant reduction in the levels of cell cycle-associated

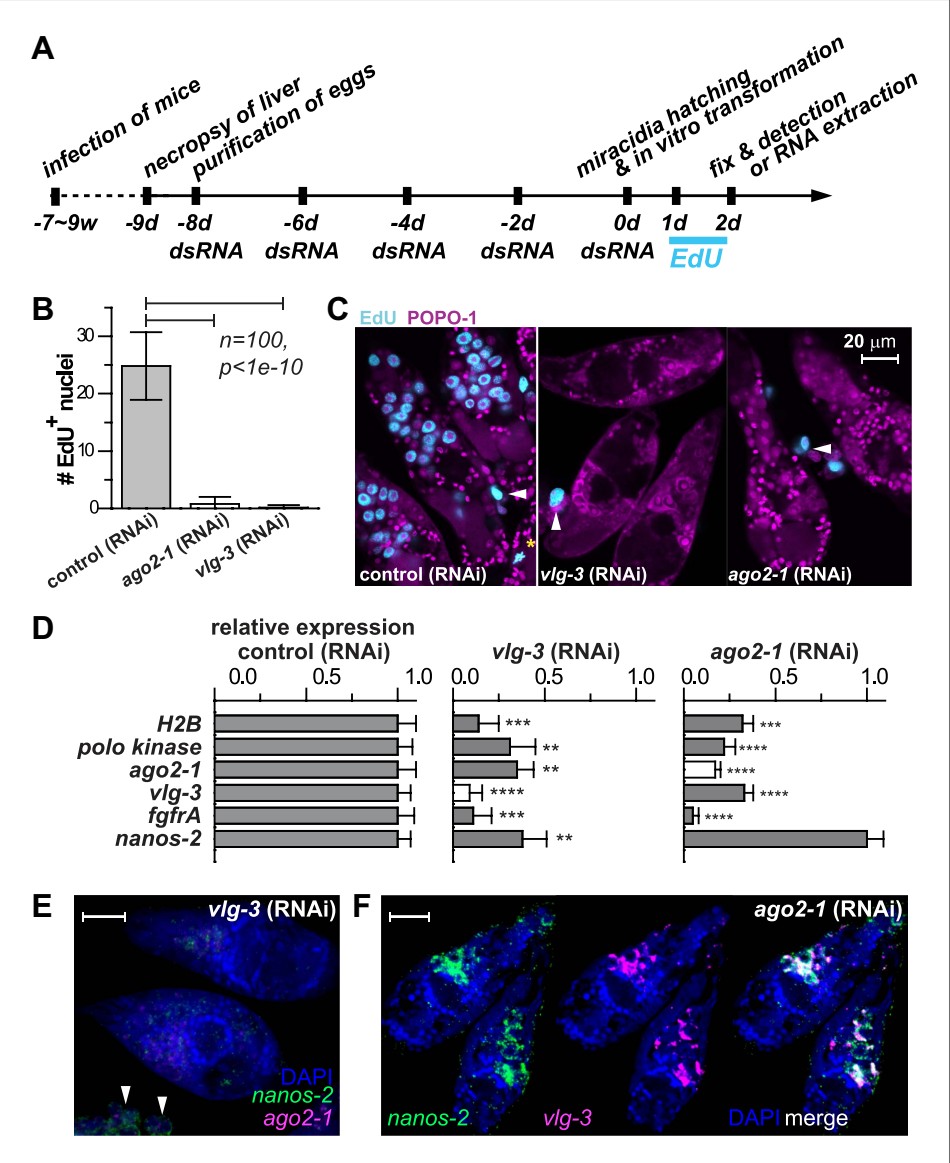

**Figure 5**. *vlg-3* and *ago2-1* are required for germinal cell maintenance and proliferation. (**A**) Timeline for RNAi experiments; EdU was added to the culture at 20–40 hr post-transformation. (**B**) Average number of EdU+ nuclei per sporocyst in control, *ago2-1* (RNAi), and *vlg-3* (RNAi) experiments. (**C**) Representative confocal stacks showing EdU incorporation after RNAi. Arrowheads indicate proliferating *Bge* cells; asterisk indicates an EdU+ mitotic cell. (**D**) Relative gene expression levels measured by qPCR for control and RNAi sporocysts, both in biological triplicate. The white bars indicate the genes targeted by RNAi. Error bars represent standard deviations. **p<0.01, ***p<0.001, ****p<0.0001 (*t*-test). (**E–F**) FISH to detect *nanos-2* and *ago2-1* expression after *vlg-3* RNAi (**E**), and *nanos-2* and *vlg-3* expression after *ago2-1* RNAi (**F**). In (**E**), non-specific binding of probes to the *Bge* cells illustrates the background levels (arrowheads). The expression patterns in control RNAi animals are unchanged. Scale bars are 20 µm.

transcripts such as *polo kinase* and *histone H2B*, as well as germinal cell-associated genes (*ago2-1*, *vlg-3*, and *fgfrA*) (*Figure 5D*). To resolve whether these effects were due to defects in cell proliferation or loss of the germinal cells, we examined the expression pattern of germinal cell markers by FISH after RNAi treatments. We observed that RNAi of *vlg-3* resulted in the loss of both *nanos-2*[+] and *nanos-2*[−] germinal cells (*Figure 5E*), suggesting that it promotes the maintenance of the entire germinal cell population. In contrast to *vlg-3,* reduction of *ago2-1* levels resulted in loss of only *nanos-2*[−] germinal cells; *nanos-2*[+] cells were still present (*Figure 5F*). However, the absence of proliferation after *ago2-1* RNAi (*Figure 5C*) suggests that the remaining *nanos-2*[+] cells fail to proliferate. These observations are consistent with qPCR quantification, in which expression of *nanos-2* was reduced after *vlg-3* RNAi but retained after *ago2-1* RNAi (*Figure 5D*).

### *fgfrA* expression and function suggest that a similar signal transduction network functions in germinal cells and adult schistosome stem cells

Noting that *ago2-1* and *nanos-2* are also expressed in somatic stem cells in adult schistosomes (*Collins et al., 2013*), we speculated that a similar network may regulate both adult stem cells and germinal cells. Indeed, *fgfrA*, which is essential for maintenance of adult stem cells, is also upregulated in sporocysts (*Figure 6A*, *Figure 2—figure supplement 1*, and *Table 1*) and its expression depends on either *ago2-1* or *vlg-3*, suggesting that *fgfrA* is present in germinal cells (*Figure 5D*). To examine the function of *fgfrA* we performed RNAi experiments. Disruption of *fgfrA* mRNA blocked germinal cell proliferation, measured by reduced EdU incorporation (*Figure 6B*) and downregulation of cell cycle-associated transcripts (*Figure 6C*). Similar to *ago2-1* RNAi parasites, *fgfrA* RNAi also resulted in reduced *ago2-1* and *vlg-3* expression levels, while having no significant effect on the expression of *nanos-2* (*Figure 6C*). These results suggest a similar role for FGF signaling in controlling the proliferation of stem cells in both larval and adult schistosomes.

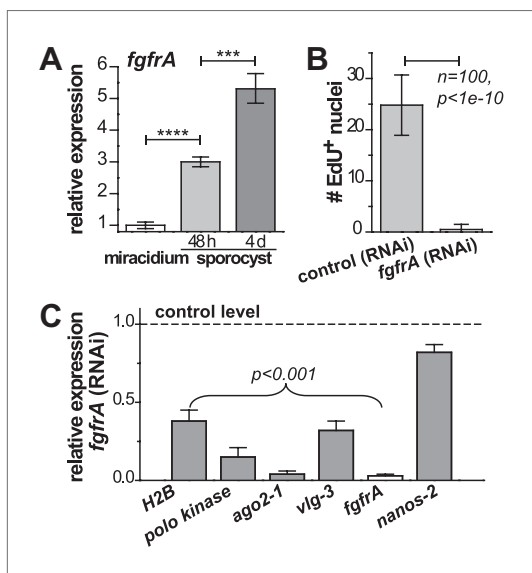

**Figure 6**. *fgfrA* is required for germinal cell proliferation. (**A**) Relative expression levels of *fgfrA* during sporocyst development with respect to the expression in miracidia, measured by qPCR. (**B**) Average number of EdU[+] nuclei per sporocyst in control and *fgfrA* (RNAi) parasites labeled at 20–40 hr post-transformation. (**C**) Relative gene expression levels measured by qPCR for control and *fgfrA* (RNAi) sporocysts. The white bar indicates efficient knockdown of *fgfrA* by RNAi. qPCR experiments were performed in biological triplicate. Error bars are standard deviations. ***p<0.001, ****p<0.0001 (*t*-test).

## Discussion

Comparison between sporocyst germinal cells described in this study and somatic stem cells recently identified in adult schistosomes (*Collins et al., 2013*) reveals significant molecular similarities, suggesting that these stem cells may persist throughout the entire schistosome life cycle. These germinal cells also exhibit many transcriptional and functional similarities with the neoblasts of free-living planarians. In planarians, PRC2 components are enriched in neoblasts, and are required for neoblast self-renewal and long-term maintenance (*Wagner et al., 2012*; *Önal et al., 2012*); disruption of *ago2* expression downregulates a specific set of miRNAs and blocks neoblast self-renewal (*Rouhana et al., 2010*; *Li et al., 2011*); *Smed-fgfr1* is expressed in a subset of planarian neoblasts; *Smed-fgfr4* (*Wagner et al., 2012*) and *Smed-bruli* (*Guo et al., 2006*) are both enriched in neoblasts; and *vasa* is required for the proliferation and expansion of neoblasts (*Rouhana et al., 2010*; *Wagner et al., 2012*). Although schistosomes do not have a true *vasa* ortholog, the *vasa*/PL10 homolog *vlg3* may have assumed a 'vasa-like' role in schistosome stem cell maintenance (*Tsai et al., 2013*). Consistent with this idea, *vasa*/PL10 homologs in monogenean flatworms, which also lack *vasa*, are essential for germline development (*Ohashi et al., 2007*). This conservation at both the cellular and molecular levels suggests an ancient role for these genes in regulating stem

cell populations in flatworms. In light of these observations, it is plausible to suggest that neoblast-driven developmental programs inherited from their free-living ancestors may have enabled the evolution of complex trematode life cycles. These pluripotent cells may help the worms adapt successfully to obligate parasitism, by enabling both rapid expansion of an infective population and long-term tissue maintenance in the hostile environments within their intermediate and definitive hosts. Thus, future studies deciphering the evolutionary relationships between various neoblast-like cell populations are essential to understand the successful transmission and pathogenesis of various parasitic flatworms.

Our data revealing the existence of two germinal cell subpopulations ($nanos$-$2^+$ and $nanos$-$2^-$ cells) are reminiscent of observations in some other trematode species, in particular *Echinostoma*. In these parasites, proliferating germinal cells are present as two morphologically distinguishable populations (*Galaktionov and Dobrovolskij, 2003*). One of these populations, smaller in cell size, was speculated to be the most 'undifferentiated' cell type, whereas the other was thought to have more restricted developmental potential and enter embryogenesis directly. Although this morphological heterogeneity was not observed in schistosomes (*Pan, 1980*), our results uncovered molecular heterogeneity in these germinal cells. The observation that germinal cells expressing $nanos$-$2$ exhibit slower cell-cycle kinetics is consistent with the conserved role of $nanos$ in lengthening the cell cycle by repressing mitotic transcripts in primordial germ cells from many animals (*Juliano et al., 2010*). Along with our observations that $nanos$-$2$ is also expressed in germ cells in schistosome adult gonads (Wang and Newmark, unpublished) as well as adult somatic stem cells (*Collins et al., 2013*), it is reasonable to expect the $nanos$-$2^+$ cells may be more totipotent or germ cell-like, whereas the $nanos$-$2^-$ germinal cells may be more primed towards somatic fates. Given recent advances (*Dvořák et al., 2010*; *Rinaldi et al., 2012*), transgenic approaches to dissect the existence of such a developmental hierarchy may be possible in the future.

Our results provide an initial molecular description of germinal cells in schistosome intramolluscan development. Based on the classic literature, the biology of these cells is quite unique: proliferative totipotent stem cells that directly undergo embryogenesis in the absence of fertilization. We find that, in spite of this unique developmental program, these cells possess a molecular signature similar to that of neoblasts in free-living flatworms, as well as stem cells from diverse organisms. This conserved molecular context opens access to understanding these cells, and may lead to strategies for intervening and blocking the transmission of the disease.

## Materials and methods

### Parasites

*S. mansoni* (strain: NMRI) life cycle stages were provided by the Biomedical Research Institute (BRI, Rockville, MD) via the NIAID Schistosomiasis Resource Center through NIH-NIAID contract no. HHSN272201000005I. To purify *S. mansoni* eggs, livers from mice (6–8 weeks post infection) were sterilized and digested in 5% clostridial collagenase (Sigma, St. Louis, MO) solution at 37°C for ~20 hr (*Mann et al., 2010*). The digested suspension was then forced through a 105-µm sieve, followed by repeated centrifugation and washing. Remaining liver tissue was removed by Percoll sucrose gradient centrifugation (*Mann et al., 2010*). Purified eggs were cultured at 37°C/5% $CO_2$ in Basch's medium 169 (*Basch, 1981*), supplemented with 10% heat-inactivated fetal bovine serum (FBS, Hyclone/Thermo, Logan, UT) and 1× antibiotic-antimycotic (Gibco, Carlsbad, CA) for up to 10 days without significantly reducing the hatching rate of miracidia.

Free-swimming miracidia were hatched in artificial pond water under bright light for 3 hr (*Samuelson et al., 1984*). Unhatched eggs (typical hatching rate ~70–80%) and empty egg shells were removed by centrifugation. Miracidia were transformed in vitro to mother sporocysts by exchanging pond water with sporocyst culture medium (*Ivanchenko et al., 1999*). The sporocyst suspensions were co-cultured with *Bge* cells at 37°C/5% $CO_2$/5% $O_2$. These conditions have been derived to maintain long-term cultures of sporocysts (*Bixler et al., 2001*).

Cercariae were obtained from *Biomphalaria glabrata* snails (Schistosomiasis Resource Center) ~5–8 weeks post infection by exposing snails to bright light at 28°C for 1–2 hr. Cercariae and miracidia were fixed in 4% formaldehyde in pond water supplemented with 0.2% Triton X-100 and 1% NP-40.

### Cryosectioning

Infected *B. glabrata* snails were fixed (4% formaldehyde in pond water with 0.2% Triton X-100 and 1% NP-40) at 4°C for at least 24 hr. Then the shell was crushed and removed, the snail tissue was equilibrated

in 30% sucrose overnight, embedded in tissue freezing medium (TBS), and cryosectioned. 30 µm sections were stained with combinations of 10 µM phalloidin conjugated to Alexa Fluor 568, 1 µM POPO-1 iodide, 5 µM SYTOX-Green (Invitrogen, Carlsbad, CA), and/or 20 µg/ml fluorescein-labeled PNA (Vector Laboratories, Burlingame, CA) (*Collins et al., 2011*). Samples were cleared in *scale* A2 mounting solution (4M Urea, 10% glycerol, 0.1% Triton X-100 in PBS) (*Hama et al., 2011*). Fluorescence images were obtained on a Zeiss LSM 710 confocal microscope, with a 63× oil immersion objective (N.A. = 1.4). Before imaging, 10 mM ascorbic acid was added freshly to the mounting solution to prevent photobleaching.

## EdU labeling and detection

In vitro transformed sporocysts were cultured with 10 µM EdU (Invitrogen, Carlsbad, CA) for the indicated time periods. Following the EdU pulse, sporocysts were fixed for 30 min at room temperature or overnight at 4°C in 4% formaldehyde in Chernin's balanced salt solution (*Chernin, 1963*) with 0.2% Triton X-100 and 1% NP-40. Fixed sporocysts were sequentially dehydrated in 50% methanol and then pure methanol. The dehydrated samples were kept at −20°C overnight, and rehydrated by exchanging methanol with 50% methanol, and then PBSTx (PBS with 0.3% Triton X-100). EdU incorporation was detected by click reaction with 25 µM Alexa Fluor azide conjugates (Invitrogen) for 20 min (*Salic and Mitchison, 2008*).

## Transcriptional profiling

Total RNA was purified from ~10,000 freshly hatched miracidia or sporocysts 48 hr post-transformation using Trizol (Invitrogen), and DNase treatment. For RNAseq, individually tagged libraries were prepared and pooled in a single lane. For each sample, ~90 million 100 bp reads were sequenced on an Illumina HiSeq2000 at the W. M. Keck Center for Comparative and Functional Genomics, and mapped to the annotated *S. mansoni* genome (version 5) (*Protasio et al., 2012*) using CLC Genomics Workbench (CLC Bio, Aarhus, Denmark). Transcriptome analyses have been submitted to NCBI under the accession number GSE48282. Comparisons between sporocyst and neoblast-enriched genes were performed using a list of 4032 transcripts whose expression was enriched in FACS-purified X1 neoblasts vs X2 neoblast progeny, and Xins differentiated cells (*Önal et al., 2012*).

For qPCR, total RNA was reverse transcribed to cDNA, and quantified on an Applied Biosystems Step One Plus station using GoTag qPCR reagents (Promega, Madison, WI). Experiments were performed with three independent biological replicates, each containing ~5000 animals. Relative expression levels were determined using the $\Delta\Delta C_t$ calculation. The relative fold changes across samples measured by RNAseq were validated by qPCR of 10 randomly picked genes, giving $R^2 \approx 0.9$. Four genes, Smp_093230 (actin-related protein 10), Smp_197220 (60S ribosomal protein L35), Smp_089880 (fad oxidoreductase), and Smp_169030 (aspartyl-tRNA synthetase) served as internal controls. These genes did not show life cycle stage-specific expression and their expression was not altered following RNAi. The primers used are listed in *Supplementary file 1E*.

## RNA FISH

The fixed, dehydrated samples were stored in methanol at −20°C for up to 2 weeks without noticeable signal deterioration. The parasites were transferred to baskets (35-µm mesh, Intavis, Koeln, Germany), rehydrated, treated with 2 µg/ml proteinase K (Invitrogen) for 3–5 min, and post-fixed for 10 min in 4% formaldehyde in PBSTx. Hybridization and detection followed the protocol developed for adult worms (*Collins et al., 2013*), except for the following modifications. The hybridization was carried out at 52°C overnight with ~500 ng/ml antisense riboprobes, synthesized with standard in vitro transcription reactions incorporating digoxigenin-12-UTP (Roche) or dinitrophenol-11-UTP (Perkin Elmer). Following stringency washes at 52°C, the samples were blocked with 1% casein and 7.5% horse serum, and incubated with anti-digoxigenin-peroxidase (1:1000; Roche, Indianapolis, IN) or anti-dinitrophenol-peroxidase (1:250; Perkin Elmer, Waltham, MA) at 4°C overnight. Detection was performed using tyramide signal amplification (TSA) with home-made tyramides (*King and Newmark, 2013*). For multi-color FISH, the peroxidase was quenched for 30 min in 0.1% sodium azide solution between sequential detections of different transcripts. Clones used for riboprobe and dsRNA synthesis were generated as described elsewhere (*Collins et al., 2010*), with oligonucleotide primers listed in *Supplementary file 1F*.

## RNAi

dsRNA was transcribed in vitro as described elsewhere (*Collins et al., 2010*). The newly purified eggs (~5000 eggs per ml) were soaked with ~20 µg/ml dsRNA for 8 days, with dsRNA freshly added every

other day (*Figure 5A*). Following hatching, the sporocysts were transformed and maintained in sporocyst medium supplemented with ~20 µg/ml of dsRNA. As a negative control, parasites were soaked with a 1.5 kbp dsRNA derived from an irrelevant bacterial sequence (*Collins et al., 2010*).

## Acknowledgements

We thank Eric Ross for sharing the reciprocal BLAST program, Ryan King, Harini Iyer, Rachel Roberts-Galbraith, Melanie Issigonis, and other Newmark lab members for experimental help, stimulating discussions, and critical reading of the manuscript. BW is supported by the Institute for Genomic Biology fellowship program. PAN is an investigator of the Howard Hughes Medical Institute.

## Additional information

### Funding

| Funder | Grant reference number | Author |
| --- | --- | --- |
| National Institutes of Health | R21 AI099642 | Phillip A Newmark |
| Howard Hughes Medical Institute | 079103 | Phillip A Newmark |

The funders had no role in study design, data collection and interpretation, or the decision to submit the work for publication.

### Author contributions

BW, JJC, Conception and design, Acquisition of data, Analysis and interpretation of data, Drafting or revising the article; PAN, Conception and design, Analysis and interpretation of data, Drafting or revising the article

## Additional files

### Supplementary files

• Supplementary file 1. (**A**) Transcripts that are upregulated in sporocysts 48 hr post in vitro transformation. (**B**) Expression of DEAD-box helicases (DBHs). (**C**) Transcripts that are conserved between planarian neoblast-enriched genes and genes that are upregulated in sporocysts 48 hr post in vitro transformation. (**D**) Orthologs between sporocyst-enriched genes and planarian neoblast-enriched transcripts. (**E**) Primers for qPCR. (**F**) Primers for cloning.

### Major dataset

The following dataset was generated:

| Author(s) | Year | Dataset title | Dataset ID and/or URL | Database, license, and accessibility information |
| --- | --- | --- | --- | --- |
| Wang B, Collins J III, Newmark P | 2013 | Data from: Functional genomic characterization of germinal cells in larval *Schistosoma mansoni* | GSE48282; http://www.ncbi.nlm.nih.gov/geo/query/acc.cgi?acc=GSE48282 | Publicly available at GEO (http://www.ncbi.nlm.nih.gov/geo/). |

**Reporting standards:** Minimum Information about a high-throughput SeQuencing Experiment—MINSEQE

The following previously published dataset was used:

| Author(s) | Year | Dataset title | Dataset ID and/or URL | Database, license, and accessibility information |
| --- | --- | --- | --- | --- |
| Önal P, Grün D, Adamidi C, Rybak A, Solana J, Mastrobuoni G, et al. | 2012 | Data from: Gene expression of pluripotency determinants is conserved between mammalian and planarian stem cells (*The EMBO Journal* (2012) 31, 2755–2769) | http://dx.doi.org/10.1038/emboj.2012.110 | Data available upon request from Önal et al. |

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
