## [Decision Letter]

Thank you for sending your work entitled “Functional genomic characterization of neoblast-like stem cells in larval *Schistosoma mansoni*” for consideration at *eLife*. Your article has been favorably evaluated by a Senior editor and 3 reviewers, one of whom is a member of our Board of Reviewing Editors.

The following individuals responsible for the peer review of your submission want to reveal their identity: Alejandro Sánchez Alvarado (Reviewing editor) and Peter Olson (peer reviewer).

The Reviewing editor and the other reviewers discussed their comments before we reached this decision, and the Reviewing editor has assembled the following comments to help you prepare a revised submission.

This groundbreaking work significantly expands our understanding of fundamental developmental processes regulating Schistosomes, a remarkably plastic and therapeutically refractory vector of a devastating human disease. The findings reported here should be of broad interest to developmental biologists, parasitologists, and tropical medicine researchers and physicians.

Because of the importance of the gene expression profile comparison between the non-parasitic flatworm (*S. mediterranea*) and Schistosome for future discovery and therapeutic research, we would like the authors to address the following:

1) Please describe how the list of 4,032 *S. mediterranea* neoblast enriched genes was obtained/selected. The authors should provide a description of the methods/criteria used to produce this list.

2) The comparison between the sporocyte sequences and the *S. mediterranea* neoblast should be carried out with somewhat higher granularity. By that we mean that reciprocal blasts should be performed to determine if the identified shared transcripts between the two species are either homologs or likely orthologs. For example, what percentage of the ∼500 genes identified shared genes between the sporocytes and the planarian neoblasts reciprocally blast to each other?

3) How many of the ∼6,000 sporocyte expressed genes would hit one of the ∼4,000 planarian genes with the blast threshold used, and how does this compare to ∼1,600 genes selected in the paper to carry out the comparison? This would help us appreciate whether the ∼30% overlap of genes reported is significant rather than what would be expected at random.

4) Some enhancement of the gene case study discussion in the text should also be considered. e.g., references to planarian *fgfr* expression, *vasa*, and *vlg* data, and PRC2 gene data; and added assessment of presence or absence for other well-studied neoblast genes such as *bruli*. It is stated “*vasa* is required for proliferation…” in comparison to the *vlg3* data – is the *vlg3* gene an ortholog of the *vasa* gene referenced or a *vlg*?

---

## [Author Response]

*1) Please describe how the list of 4,032* S. mediterranea *neoblast enriched genes was obtained/selected. The authors should provide a description of the methods/criteria used to produce this list*.

As now clarified in the text, these genes were obtained from a planarian neoblast transcriptome constructed by Rajewsky and coworkers (34). We have modified the main text to read: “… we compared our dataset with transcripts enriched in FACS-purified planarian neoblasts (34) to identify putative germinal cell-specific transcripts.”

We have also modified the Materials and methods: “Comparisons between sporocyst and neoblast-enriched genes were performed using a list of 4032 transcripts whose expression was enriched in FACS-purified neoblasts vs. X2 neoblast progeny, and Xins differentiated cells (34).”

*2) The comparison between the sporocyte sequences and the* S. mediterranea *neoblast should be carried out with somewhat higher granularity. By that we mean that reciprocal blasts should be performed to determine if the identified shared transcripts between the two species are either homologs or likely orthologs. For example, what percentage of the ∼500 genes identified shared genes between the sporocytes and the planarian neoblasts reciprocally blast to each other*?

We have carried out a full-genome reciprocal BLAST, and identified 331 orthologs that are upregulated in both sporocysts and FACS-sorted neoblasts. These genes are now highlighted in the new Figure 2, listed in a new Supplementary table 4 in [Supplementary-material SD1-data], and they are discussed in the text:

“To better define the similarity between these datasets, we identified a total of 4749 1:1 schistosome-to-planarian orthologs using reciprocal BLAST comparisons (e-value<e^-10^) (Figure 2). […] We examined the overlap between neoblast-enriched transcripts and sporocyst-enriched genes and identified 331 orthologs (20% of 1662 genes that are upregulated in sporocysts).”

*3) How many of the ∼6,000 sporocyte expressed genes would hit one of the ∼4,000 planarian genes with the blast threshold used, and how does this compare to ∼1,600 genes selected in the paper to carry out the comparison? This would help us appreciate whether the ∼30% overlap of genes reported is significant rather than what would be expected at random*.

The miracidia-to-sporocyst transition is associated with not just an expansion of germinal cells, but also numerous physiological and anatomical changes. Thus, we initially performed the comparative analyses between sporocysts and planarian neoblasts as a means to discover candidates for further studies. Now as the reviewers have suggested, we have included in the text a more quantitative comparison:

“…1579 orthologs were enriched in planarian neoblasts; quite interestingly, 1525 of them (96.5%) were expressed in sporocysts as well (RPKM in sporocysts >1). We examined the overlap between neoblast-enriched transcripts and sporocyst-enriched genes and identified 331 orthologs (20% of 1662 genes that are upregulated in sporocysts)…”

By contrast, 12% (1248/10125) of the remaining schistosome transcripts have orthologs in the planarian neoblast transcriptome, but further inspection suggests this background overlap arises mostly from house-keeping genes. While this enrichment in our dataset for genes with orthologs in planarian neoblasts is confirmative, we also noted this in the text:

“Given the differences in how these datasets were generated (i.e., FACS-purified cells from planarians vs. whole schistosome sporocysts), together with fact that the miracidium-to-sporocyst transformation is expected to result in transcriptional changes independent of germinal cell number, this degree of overlap seems likely to underestimate the similarity between the transcriptomes of germinal cells and planarian neoblasts.”

Examining this overlapping gene set we were able to identify many conserved factors associated with stem cell maintenance and germ cell development in diverse organisms (Table 1), suggesting this overlap bears biological significance.

*4) Some enhancement of the gene case study discussion in the text should also be considered. e.g., references to planarian* fgfr *expression,* vasa*, and* vlg *data, and PRC2 gene data; and added assessment of presence or absence for other well-studied neoblast genes such as* bruli*. It is stated “*vasa *is required for proliferation…” in comparison to the* vlg3 *data – is the* vlg3 *gene an ortholog of the* vasa *gene referenced or a vlg*?

To enhance the discussion of the case studies as suggested, we have modified the text and included five more references. This revised section begins:

“In addition to components of cell cycle and DNA repair machinery, examination of these overlapping gene sets identified many conserved factors associated with stem cell maintenance and germ cell development in diverse organisms…”

We have also modified the Discussion (section beginning/ending): “In planarians, PRC2 components are enriched in neoblasts, and are required for neoblast self-renewal and long-term maintenance (51; 34); […] Consistent with this idea, *vasa*/PL10 homologs in monogenean flatworms, which also lack *vasa*, are essential for germline development (32).”